# Monoterpene Synthase Genes and Monoterpene Profiles in *Pinus nigra* subsp. *laricio*

**DOI:** 10.3390/plants11030449

**Published:** 2022-02-06

**Authors:** Enrica Alicandri, Stefano Covino, Bartolomeo Sebastiani, Anna Rita Paolacci, Maurizio Badiani, Agostino Sorgonà, Mario Ciaffi

**Affiliations:** 1Dipartimento di Agraria, Università Mediterranea di Reggio Calabria, Loc. Feo di Vito, I-89129 Reggio Calabria, Italy; e.alicandri@gmail.com (E.A.); mbadiani@unirc.it (M.B.); asorgona@unirc.it (A.S.); 2Dipartimento per la Innovazione nei Sistemi Biologici, Agroalimentari e Forestali, Università della Tuscia, Via S. Camillo De Lellis, s.n.c, I-01100 Viterbo, Italy; stefano.covino80@gmail.com (S.C.); arpaolacci@unitus.it (A.R.P.); 3Dipartimento di Chimica, Biologia e Biotecnologie, Università di Perugia, Via Elce di Sotto, 8, I-06123 Perugia, Italy; bartolomeo.sebastiani@unipg.it

**Keywords:** monoterpenes, monoterpene synthase genes, *Pinus nigra* subsp. *laricio*, genomic organization, pine oleoresin, terpene volatile compounds, β-phellandrene, α-pinene, β-pinene

## Abstract

In the present study, we carried out a quantitative analysis of the monoterpenes composition in different tissues of the non-model conifer *Pinus nigra* J.F. Arnold subsp. *laricio* Palib. ex Maire (*P. laricio*, in short). All the *P. laricio* tissues examined showed the presence of the same fourteen monoterpenes, among which the most abundant were β-phellandrene, α-pinene, and β-pinene, whose distribution was markedly tissue-specific. In parallel, from the same plant tissues, we isolated seven full-length cDNA transcripts coding for as many monoterpene synthases, each of which was found to be attributable to one of the seven phylogenetic groups in which the d1-clade of the canonical classification of plants’ terpene synthases can be subdivided. The amino acid sequences deduced from the above cDNA transcripts allowed to predict their putative involvement in the biosynthesis of five of the monoterpenes identified. Transcripts profiling revealed a differential gene expression across the different tissues examined, and was found to be consistent with the corresponding metabolites profiles. The genomic organization of the seven isolated monoterpene synthase genes was also determined.

## 1. Introduction

### 1.1. Terpenoids: Basic Definitions and Roles in Conifers

Terpenoids, also referred to as terpenes or isoprenoids, compose the biggest and most diversified class of chemical substances discovered in plants, encompassing over 40,000 individual compounds [1,2,3]. The evolutionary success of the terpenoid metabolites has largely depended on the flexibility of their building molecules of various sizes. Indeed, terpenoids, arising from the two basic five-carbon (C_5_) isoprenoid units, isopentenyl diphosphate, and its isomer, dimethylallyl diphosphate, can be categorized as hemiterpenoids (C_5_), monoterpenoids (C_10_), sesquiterpenoids (C_15_), diterpenoids (C_20_), triterpenoid (C_30_), tetraterpenoid (C_40_), or polyterpenoids (C_5n_), based on the number of C_5_ units they contain [4].

The production of terpenoids in conifer species, both in the form of oleoresin and emitted as volatile compounds, plays a significant role in the physical and chemical defence responses against herbivores and pathogens [5,6]. Oleoresin, whose main components are mono- and diterpenes (including diterpene resin acids, DRAs), with lower quantities of sesquiterpenes, accumulates in specialized anatomical structures, mainly resin ducts, which are abundantly present in different tissues of conifer trees, such as their bark, wood, and needles [5]. In case of wounding, the resin spreads out from the ducts to reach the damaged area and acts as a physical and chemical weapon against invading organisms [5,7]. Conifer foliage also emits terpenes originating from two sources: those stored in the needle resin ducts and those directly emitted from mesophyll cells [5]. These volatile terpenes, mainly mono- and sesquiterpenes, can play a significant role in plant-to-environment interactions, acting as signals to stimulate defence responses in nearby plants or in healthy tissues of the same plant or in attracting natural enemies of pathogens and herbivores [5,8,9].

### 1.2. Ecophysiology and Biotechnology of Conifer Monoterpenes

Conifer monoterpenes, as major components of the oleoresin and of the emitted volatile terpene fraction, are largely involved in the constitutive and induced defence responses of conifer trees to insects and fungal diseases [5,10]. For instance, Nicole et al. [11] showed that in Norway spruce trees (*Picea abies*) attacked by the white pine weevil (*Pissodes strobi*), a harmful forest pest found across Canada and the northern USA causing permanent stem and crown deformations, a rapid increase in the monoterpene concentration was found, with different levels measured in the needles and bark. In addition, traumatic resin duct formation and resin accumulation were observed, indicating not just a chemical plant response, but also a physical one. Another study conducted by Hall et al. [12] on Sitka spruce (*Picea sitchensis*) revealed that when under attack by the white pine weevil, resistant trees produced much higher levels of (+)-3-carene than susceptible ones, in which only trace amounts were found of such a monoterpene.

Additionally, in the case of an interaction with pathogenic fungi, a general increase in monoterpene levels was observed. Pollastrini et al. [13] showed that inoculation with *Heterobasidion annosum* or *H. irregular*, the causal agents of annosus root rot, caused an increase in the levels of monoterpenes, such as (−)-β-pinene, in cortical tissue samples from *Pinus pinea*, suggesting a direct role of these molecules in the defence mechanisms.

Monoterpenes are also involved in the constitutive and induced defence response of lodgepole pine (*Pinus contorta*) against the mountain pine beetle (MPB) (*Dendroctonus ponderosae* Hopkins) and its associated fungus, *Grosmannia clavigera*, although their effectiveness against the MPB is variable and may depend on the beetle population density [5,14]. Indeed, lodgepole pine trees respond with an induced accumulation of monoterpenes after a MPB attack or after inoculation with *G. clavigera*, and lodgepole pine trees with high levels of monoterpenes experienced a decreased frequency of attacks at a low beetle population density [14]. However, terpene defences that are effective in protecting lodgepole pine trees against the MPB at low levels of infestation during the endemic phase may become inefficient under epidemic conditions [15]. Moreover, adult MPBs employ pine monoterpenes as signal molecules to find suitable host trees and as precursors for their pheromone biosynthesis [5,14].

In addition to a chemical and physical defence, constitutive and induced conifer terpenes, mainly hemi- and monoterpenes, can fulfil a physiological and ecological role in the protection against abiotic stresses, such as drought, salinity, high and low temperatures, and air and soil pollution, reviewed in [10]. Moreover, their emission from conifer trees plays a considerable role in the global carbon cycle and atmospheric processes (e.g., in the cycle of photochemical production/destruction of ozone, particle formation, and aerosols) [16,17]. Many stress conditions cause a rapid rise in the formation of reactive oxygen species, resulting in oxidative stress and the activation of signalling pathways that lead to metabolic reprogramming [18]. Several studies showed that some terpenes exhibit considerable antioxidant activity [19,20], suggesting that they may have a role in preventing oxidative stress induced by abiotic stressors [10].

While the constitutive and induced terpene-related defence mechanisms are thought to have contributed to the evolutionary success of conifers in forest ecosystems worldwide, the increased pressure from the combined and related effects of climate change and the expansion of conifer pests and diseases may challenge the effectiveness of these defence systems [5].

Because many monoterpenes are volatile molecules, some of them, including ocimene, myrcene, limonene, linalool, camphene and pinene, just to mention a few of those found in conifers, are well-known fragrances found in essential oils, and are used in perfumes, cosmetics, and cleaning products [3]. Moreover, several conifer monoterpenes, such as bornyl acetate, camphene, and limonene, are used as food flavours and food additives [21]. Finally, yet also importantly, a few conifer monoterpenes have been claimed to exhibit antimicrobial, antiviral, or antitumor activities [22,23,24].

### 1.3. Biosynthesis of Monoterpenes in the Pinaceae: Genes and Enzymes

Monoterpenes are a class of terpenes that consist of two isoprene units and have the molecular formula C_10_H_16_. Monoterpenes may be linear (acyclic) or contain rings (monocyclic and bicyclic). Modified terpenes, such as those containing oxygen functionality or missing a methyl group, are called monoterpenoids.

The biosynthesis of monoterpenes results from the activity of a specialized group of terpene synthases (TPSs), namely, monoterpene synthases (MTPSs), which use geranyl diphosphate as a substrate to form, typically in a stereo-specific fashion, acyclic, monocyclic, or bicyclic compounds. All the known MTPSs belong to the so called “class-I” TPSs, hosting only one active site, which is located in the C-terminal α-domain. Such α-domain contains two metal-binding amino acidic motifs, namely, the highly conserved “DDXXD” and the less conserved “NSE/DTE” one. Most MTPSs possess an obvious N-terminal transit peptide, which is needed for their import into plastids. Such a transit peptide is removed from the mature MTPS upstream of the “RR(X8)W” motif, which in turn is important for the catalysis of monoterpene cyclization (see [6] and references therein).

The abundance and diversity of the different monoterpenes in conifers [10], and their roles in conifer–herbivore interactions, prompted the cloning and characterization of MTPSs in several conifer species, reviewed in [5,6]. For instance, the white spruce (*Picea glauca*) genome annotation identified 24 MTPS genes [25], but only 4 of them have been functionally characterised, i.e., heterologously expressed in bacterial/yeast systems and tested in vitro with their potential terpenoids substrates, including (−)-α-pinene synthase, (−)-β-pinene synthase, (−)-linalool synthase, and 1,8-cineole synthase [26]. Another study allowed the cloning of nine different jack pine (*Pinus banksiana*) and eight different lodgepole pine (*Pinus contorta*) MTPS full-length cDNAs [14]. The functional characterization of the 17 different MTPSs and monoterpene profiles in different tissues of lodgepole pine and jack pine trees identified orthologous sets of enzymes that contribute to the biosynthesis of oleoresin monoterpenes such as (−)-β phellandrene, (−)-β-pinene, (+)-α-pinene, (−)-α-pinene, and (+)-3-carene [14].

### 1.4. Aims and Background of the Present Work

It became apparent from the above-mentioned work that most of the existing knowledge concerning the genetics and metabolism of monoterpenes in conifers was obtained from model Pinaceae species, for which large transcriptomic and genomic resources are available. In previous works of ours [6,27,28], we began to gain insight into the ecological and functional roles of the terpenes produced by the non-model conifer *Pinus nigra* J.F.Arnold subsp. *laricio* Palib. ex Maire, *P. laricio* for short in the following, vernacular name Calabrian pine, one of the six subspecies of *P. nigra* (black pine) and an insofar completely neglected species under such respect. In terms of natural distribution, black pine is one of the most widely distributed conifers over the whole Mediterranean basin, and its *laricio* subspecies is considered endemic of Southern Italy, especially of Calabria, where it is a basic component of the forest landscape, playing key roles not only in soil conservation and watershed protection, but also in the local forest economy [29].

In one of our previous works on *P. laricio* [28], five different plant tissues were analysed in terms of the oleoresin diterpenoids composition, and, in parallel, the isolation, characterization, and expression analyses of diterpene synthase (DTPS) genes were carried out in the same plant material. To broaden the characterization of the terpene oleoresins in this non-model conifer species, the same combined biochemical and genetic approach is adopted in the present work, in which, again, for the first time to the best of our knowledge, we use gas chromatography–mass spectrometry (GC–MS) to determine the monoterpenes profiles in five different tissues of *P. laricio*. In this same subspecies, in addition, here, we report about the isolation of full-length (FL) cDNAs and the corresponding genomic sequences encoding for six MTPSs and one hemi-TPS, obtained by using a strategy based on the phylogeny of available MTPSs from different *Pinus* species. The isolation of *P. laricio MTPS* genes allowed a tissue-specific gene expression analysis to compare with the corresponding GC–MS monoterpene profiles. The results presented here on *P. laricio* are compared to, and discussed against, those obtained in the aforementioned twin study of ours [28], focusing on the diterpenoid components of oleoresin from the same species, in an attempt to expand and consolidate the basic knowledge on this non-model conifer, possibly leading to applications of interest in bio-based technology.

## 2. Results and Discussion

### 2.1. The Tissue-Specific and Species-Specific Monoterpene Metabolites in the Pinaceae

The diversity of the monoterpene composition was evaluated in five different tissues, namely, young (YN) and mature (MN) needles, leader and interwhorl stems (LS and IS, respectively), and roots (R), obtained from three-year-old *P. laricio* saplings. All the tissues examined contained the same set of fourteen monoterpene compounds, namely, bornyl acetate, camphene, δ-3-carene, α-fenchene, limonene, myrcene, β-phellandrene, α- and β-pinene, sabinene, α-terpineol, terpinolene, α-thujene, and tricyclene, whose typical elution order under the adopted GC conditions is reported in Appendix A, and whose chemical structures are shown in Appendix A. A comparison with homologous tissues obtained from coeval saplings of *Pinus banksiana* Lamb. (jack pine) and *Pinus contorta* Douglas (lodgepole pine) [14] showed a remarkably similar monoterpene profiling in *P. laricio*, albeit not identical, in that the latter contained α-thujene and α-fenchene in addition, but was devoid of α-terpinene, γ-terpinene, linalool, and terpin-4-ol.

Quantitatively speaking, Figure 1A shows that the highest content of total monoterpenes was found in the LS tissue, while decreasing amounts were observed in R, IS, and MN tissues, with the lowest concentration detected in YN.

Three of the five tissues, namely, YN, MN, and IS, contained 31–44% β-phellandrene as the most abundant monoterpene (Figure 1B,C,E), whereas α-pinene (approximately 44% of the total) and β-pinene (about 51% of the total) were found to be the most abundant monoterpenes in LS (Figure 1D) and R (Figure 1F), respectively. These findings confirmed the results obtained by Hall et al. [14] on the same tissues of three-year-old *P. contorta* seedlings, with the exception that, in the leader stem of lodgepole pine, the predominant monoterpene was β-phellandrene (more than 50% of the total), with a very low amount of α-pinene (approximately 3%).

Besides β-phellandrene, the most abundant monoterpenes found in the needles (both YN and MN) were α-pinene (26–27% of the total), β-pinene (7–17%), and limonene (6–14%), with lower amounts (less than 3% each) of the remaining ten compounds (Figure 1B,C). Conversely, the root and the stem (both LS and IS) revealed a more fragmented monoterpenes composition (Figure 1D–F). For instance, the root, in addition to β-pinene (more than half of the total monoterpene fraction), contained approximately 12% each of β-phellandrene and myrcene, 6% each of α-pinene and camphene, 4% each of α-thujene and limonene, and less than 1% of each of the remaining seven monoterpenes (Figure 1F). In this context, it is worth nothing that δ-3-carene, which was previously found to be one of the most abundant monoterpenes in *P. contorta* roots (more than the 30% of the total) [14], was found to be present at very low levels in the same tissue of *P. laricio* (0.036% of the total). On the other hand, monoterpene profiles of *P. laricio* LS and IS tissues contained 13–44% α-pinene, 11–31% β-phellandrene, 13–17% α-thujene, 9–13% β-pinene, 7–8% δ-3-carene, 3–8% myrcene, and less than 3% of the other eight identified monoterpene compounds (Figure 1E,F). In these last two tissues, the amount of α-thujene was remarkable, which, as stated before, was not identified in *P. banksiana* and *P. contorta* [14]. This bicyclic compound was found to represent 1–10% of the total monoterpene fraction in several widespread American conifers, such as *Pinus strobus* (eastern white pine), *Tsuga canadensis* (eastern hemlock), *Thuja occidentalis* (eastern white cedar), and *Juniperus* spp. (juniper) [30]. At the same time, in these same tissues, it is worth nothing the relatively high amount of δ-3-carene, which, contrary to what was observed for the roots, was comparable to that determined in the leader and interwhorl stem tissues from *P. contorta* and *P. banksiana* [14].

As Figure 1 indicates, the needles, particularly the young ones, were the tissues in which, comparatively speaking, the lowest accumulation of monoterpenes occurred, which could be of interest to understand the possible functional and ecological basis of such a circumstance. One possible explanation could be that, in the needles, monoterpenes are not only components of oleoresin, but are also actively emitted as volatile compounds to execute the well-known array of physiological and ecological functions in plant-to-environment interactions (see Introduction). As a matter of fact, in a previous study of ours conducted on adult individuals of *P. laricio* thriving in the same natural context of the present study [27], we adopted the “headspace” GC–MS approach to collect volatile terpenoids emitted from vegetation, and we found that, together with a ten of different sesquiterpenes, at least half of the same monoterpenes reported in Figure 1, namely, bornyl acetate, limonene, myrcene, β-phellandrene, α- and β-pinene, and α-terpineol, were persistently and consistently present in the blend of volatiles released by the young needles throughout their growing season, with β-phellandrene and α- and β-pinene being the major components on a comparative basis.

Taken together, the above results suggested that the quantitatively profile of monoterpenes in *P. laricio* is remarkably tissue-specific, and the perusal of the available literature indicated a species specificity as well; thus, confirming our previous results on the diterpenoids from the same conifer species [28]. As it was shown by several studies [10,30,31], there is a notable variation in the monoterpene composition between different genera and species of conifer trees. Moreover, even closely related species may differ significantly with regard to the quality and quantity of monoterpene compounds they produce, as observed in the *Pinus* genus [10,14,32,33].

Another aspect of potential interest in the non-model conifer species studied here, in view of its possible physiological, ecological, and technological implications, concerns the reciprocal quantitative relationships among the major terpenoids components of oleoresin, as well as the developmental and environmental factors driving the differential distribution of them in plant tissues. In our aforementioned twin study [28], we analysed quantitatively, in the identical plant material studied here and by adopting an essentially similar GC–MS approach, the diterpenoid components of oleoresin in the form of diterpene resin acids (DRAs), i.e., the stable and functional conformers deriving from the combined action of DTPSs and of cytochrome P450 monooxygenases [1]. By comparing the results obtained in the two experiments (Figure 1 and [28]) on a DW basis, total monoterpenes as a whole in *P. laricio* are about six times more abundant than total DRAs. Upon dissecting such cumulative data at the level of each specific tissue, however, it can be found that while such an approx. 6:1 ratio among monoterpenes and DRAs is maintained in the leader stem and in the root, it is otherwise substantially lower, i.e., approx. 2–3:1, in the interwhorl stem, but it is much higher, as much as 60:1 and 120:1, in the mature and young needles, respectively. Although a much more rigorous analysis would be needed, and albeit, the third macro-component, namely, sesquiterpenes, was not considered here, the above considerations confirm that monoterpenes are the most abundant terpenoids in conifer oleoresin [5] and seem to suggest that the relative abundance of the different terpenoids categories might be tissue-specific. While the possible functional significance, if any, of such tissue specificity awaits and deserves further studies, it can be hypothesized that the massive prevalence of monoterpenes over DRAs in needles, and especially in the young ones, might be associated with the ability of the former terpenoids, but not of the latter, to act as volatile signals for the chemical communication with the plant’s environment (see above), by using the most obvious and pervious communication channel between the plant and the surrounding atmosphere, i.e., the stomata.

### 2.2. A Phylogeny-Based Approach for Isolating Partial and Full-Length cDNAs Coding for Monoterpene Synthases in P. laricio

Based on the same strategy we previously used for isolating *DTPS* genes in the same non-model conifer species [28], a phylogeny-based approach was adopted to isolate *MTPS* cDNA sequences potentially involved in the synthesis of the monoterpenes identified in the different tissue types of *P. laricio* (see Section 2.1, above). In practice, the PCR amplification of cDNA sequences was conducted by utilizing specific primer pairs designed on conserved regions of available *MTPS*s from *Pinus* species belonging to different phylogenetic groups.

Phylogenetic relationships concerning conifer MTPS were drawn from a broader study [6], in which we conducted an exhaustive in silico search to discover all possible FL *TPS*s for primary and specialized metabolisms in several *Pinus* species. As far as the present work is concerned, BLAST searches used as queries selected gymnosperm MTPSs enabled us to identify 74 FL sequences implicated in the synthesis of hemi- and monoterpenes in pine species [6]. Indeed, the deduced amino acid sequences from 42 cDNA sequences out of 74, belonging to 18 *Pinus* species, were predicted to synthetize 2-methyl-3-buten-2-ol (MBO), a C_5_ alcohol related to isoprene from a structural and biosynthetic point of view, since both are derived from the common precursor dimethylallyl diphosphate [34] (referred to as MBO synthases, MBOSs, in the following). Because the predicted amino acid sequences of the 42 MBOSs showed a high level of homology among each other (93–99% identity), only five of them were used in the phylogenetic analysis, together with the remaining 32 FL cDNA retrieved, whose deduced amino acids sequences were predicted to belong to proper MTPSs [6]. The resulting phylogenetic tree showed that all the 37 pine MBOS/MTPS sequences clustered together into the d1 clade of TPS, which includes seven well-supported major groups, denoted as 1–7. The first group contained the five selected MBOSs, while each of the remaining six groups contained MTPS proteins thought to be functionally related among each other [6].

In each of the seven groups, we aligned the deduced amino acid and nucleotide sequences contained therein (Appendix A) to reveal group-specific highly conserved regions, which were then used to design group-specific primers, shown in the upper panel of Appendix A, for the isolation of the partial transcripts of orthologous genes in *P. laricio*.

By using the above strategy, we isolated and sequenced partial *MBOS/MTPS* transcripts of putative orthologous genes in *P. laricio*, one for each of the groups 1–7, which were then utilised as templates for the isolation of as many full-length cDNAs by carrying out 5′ and 3′ RACE (Rapid Amplification of cDNA Ends) extensions. The primer sequences used for the 5′ and 3′ RACE extensions are listed in the middle panel of Appendix A.

In the case of the partial *MTPS* transcripts from groups two, four, and seven, two slightly different sequences were identified among the three clones analysed for each cDNA fragment due to nucleotide substitutions, the majority of which were synonymous, i.e., not changing the identity of the amino acid encoded by the nucleotides triplet, on a background of otherwise high level of sequence identity among each other (over 96%). These slightly different *MTPS* transcripts, already observed in other *Pinus* species [14] and reminiscent of an analogous observation we previously determined during the isolation of *DTPS* transcripts [28], might derive from alleles of the same gene or from duplicated copies of the same gene, implying that *P. laricio* may contain many more *MTPS* closely related genes belonging to each phylogenetic group, a possibility which deserves further studies. However, among the three sequenced clones for the 5′ and 3′ RACE products of each of the three partial MTPS transcripts, we identified the same sequences which were identical to the overlapping 5′ and 3′ regions of two of the three sequenced cDNA products for each of the three genes belonging to groups two, four, and seven. These results indicated that we were able to identify at least one of the two putative FL transcripts for groups two, four, and seven. Therefore, the seven assembled unique FL cDNAs isolated from *P. laricio* and denoted as *Pnl MBOS1/Pnl MTPS2-7*, each belonging to one of the seven TPS-d1 groups, contained open reading frames (ORFs) of 1845, 1881, 1866, 1887, 1911, 1866, and 1890 bp, respectively, and were predicted to encode proteins of 614, 626, 624, 628, 636, 622, and 629 aa (Figure 2).

The FL cDNA sequences of the *Pnl MBOS1/Pnl MTPS2-7* genes were deposited in the GenBank database under the accession numbers from OL689404 to OL689410.

### 2.3. Sequence-Based Analysis of the Predicted MBOS and MTPS Proteins in P. laricio

The deduced amino acid sequences of the seven FL cDNAs obtained from *P. laricio* were found to possess highly conserved and distinctive regions of plant MTPSs (Figure 2). First, each of the seven predicted proteins featured a potential transit peptide, ranging in length from 50 (Pnl MBOS1) to 67 amino acids (Pnl MTPS4) (Figure 2), which has been predicted to facilitate the import of mature proteins into plastids. Such putative transit peptides were located just before a conserved RR(X_8_)W motif (Figure 2), which is thought to be required for the catalysis of monoterpene cyclization [6,35,36]. Furthermore, all seven predicted proteins showed a conserved Asp-rich domain, namely, DDxxD, which is thought to be responsible for class I activity and coordinating substrate binding through the formation of divalent cation salt bridges [37,38].

As shown in Figure 3, phylogenetic relationships were found among the 37 pine MBOS/MTPS sequences retrieved from the NCBI database (Appendix A) and the 7 MBOS/MTPS sequences obtained from *P. laricio*; thus, confirming the validity of the approach used for their isolation. In particular Pnl MBOS1 was clustered with the five selected pine MBOSs in phylogenetic group one, of which only Psab MBOS1 was previously found to produce both MBO and isoprene in a 90:1 ratio [34]. The high amino acid sequence identity detected among Pnl MBOS1 and Psab MBOS1 (about 95%) suggested that we isolated an FL transcript from a putative MBOS orthologous gene in *P. laricio*.

Pnl MTPS2 from *P. laricio* was found to cluster in phylogenetic group two, together with two proteins, namely, Pc MTPS6 and Pb MTPS5, which were previously found to produce α-terpineol as their major product [14].

Pnl MTPS3 from *P. laricio* was closely related (92–93% protein sequence identity) to three proteins, namely, Pb MTPS 6–7 and Pc MTPS4, assigned to phylogenetic group three, which were previously shown to produce (+)-3-carene as their major product [14]. The most remarkable difference among the four proteins was a deletion of 15 bp in the nucleotide sequence of Pnl MTPS 3, which determines the loss of five amino acids in its C-terminal region (Appendix A).

Pnl MTPS4 was found to be closely related (93–96% sequence identity) to seven MTPSs from four *Pinus* species belonging to phylogenetic group four, namely, Pc MTPS 2 and Pb MTPS 2–4, which were reported to produce (−)-β-pinene as their major product as well as (−)-α-pinene, although in comparatively lower amounts [14], and Pt MTPS2, which was shown to form (−)-α-terpineol, but neither (−)-β-pinene nor (−)-α-pinene [39].

Pnl MTPS5 from *P. laricio* was clustered in phylogenetic group five together with ten putative α-pinene synthases, among which only three, namely, Pt MTPS1, Pc MTPS1, and Pb MTPS1, have been functionally characterized as producing (−)-α-pinene as their dominant product [14,39]. Pnl MTPS5, although highly similar to the three functionally characterised α-pinene synthases (94–95% protein sequence identity), showed an insertion of six aa in its N-terminal region, not present in any members of phylogenetic group five (Appendix A).

Pnl MTPS6 from *P. laricio* was found to cluster in phylogenetic group six (Figure 3), together with four MTPSs, namely, Pc MTPS7-9 and Pb MTPS11, which were functionally characterized by Hall et al. [14]. These authors showed that Pc MTPS8, Pc MTPS9, and Pb MTPS11 formed (−)-β-phellandrene as their major product, whereas Pc MTPS7, although showing a 95% identity with the aforementioned proteins, predominantly produced (−)-camphene and (+)-α-pinene, together with other monoterpene products to a lesser extent.

Finally, Pnl MTPS7 from our conifer species was clustered in phylogenetic group seven (Figure 3) together with six MTPSs from four pine species, of which Pt MTPS3, Pb MTPS8, and Pc MTPS5 were functionally characterized as forming (+)-α-pinene as their dominant product [14,39].

In summary, placing the phylogenetic analysis reported in Figure 3 on the background of the available literature led us to hypothesize that, on the basis of their predicted protein sequences, the seven FL *MBOS/MTPS* cDNAs isolated from *P. laricio* could be involved in the biosynthesis of 2-methyl-3-buten-2-ol (Pnl MBOS1), α-terpineol (Pnl MTPS2), (+)-3-carene (Pnl MTPS3), (−)-β-pinene/(−)-α-pinene (Pnl MTPS4 and Pnl MTPS5), and (−)-β-phellandrene/(−)-camphene/(+)-α-pinene (Pnl MTPS6 and Pnl MTPS7). As a matter of fact, except for MBO, which probably escaped the extraction methodology used here probably because of its volatile nature [40], all the aforementioned monoterpenes were actually found to be present in all the tissues of *P. laricio* examined, some of which, such as β-phellandrene, α-pinene, and β-pinene, were found even if in comparatively conspicuous amounts (Figure 1; see also Section 2.5, below). It is important to recall, however, that predicting potential MTPS functions solely based on sequence homology could not always be reliable, as proven by the apparent lack of structure–function correlation previously reported for some members of the TPS-d1 clade (see above). For this reason, a functional characterization of the isolated FL transcripts through the expression of recombinant proteins in bacterial or yeast systems and in vitro enzyme assays would be crucial to decipher the actual functional roles of *P. laricio* MTPSs, regardless of their sequence homology with MTPSs from other conifers.

### 2.4. Genomic Organization of MBO/Monoterpene Synthases in P. laricio on the Background of MTPS Functional Evolution

The genomic sequences encoding the seven MBOS and MTPS proteins in *P. laricio* were amplified by using primers designed for the 5′ and 3′ termini of the coding region of the corresponding cDNAs (Appendix A) and the genomic DNA extracted from *P. laricio* needles as the template. In all cases, a single PCR product larger than the corresponding cDNA was obtained. Three clones for each of the seven genomic fragments were partially sequenced at their ends using universal primers; once the correspondence of each cDNA with its parent genomic fragment was confirmed, a single clone for each gene was chosen and both strands were completely sequenced using internal primers designed based on the corresponding cDNA.

The lengths of the genomic sequences encompassing the ORFs of the seven genes were: *Pnl MTPS1*: 2988 bp; *Pnl MTPS2*: 3313bp; *Pnl MTPS3*: 2978 bp; *Pnl MTPS4*: 3386 bp; *Pnl MTPS5*: 2854 bp; *Pnl MTPS6*: 2893 bp; *Pnl MTPS7*: 3132 bp. These genomic sequences were deposited in the GenBank database under the accession numbers from OL689411 to OL689417.

A sequence alignment showed an almost perfect match among the cDNAs and the corresponding exonic regions of the genomic sequences, allowing a reliable determination of the exon/intron structure of each gene. All the seven genomic sequences were found to contain 10 exons and 9 introns (Table 1), consistent with the previously characterized genomic sequences of conifer MTPSs [12,41,42]. Moreover, the genomic structural characteristics of the seven *MBOS/MTPS* genes from *P. laricio* were found to be highly conserved, in terms of intron placements, the exon number and sizes, and the location of RR(X_8_)W and class-I DDxxD amino acid motifs (Figure 4). The intron sizes were generally small (about 70–200 nt), except for the last intron of *Pnl MTPS4*, encompassing 469 bp (Table 1). In addition, the introns of the seven *MBOS/MTPS* genes were AT-rich, with repetitive sequences rich in T (3–10 mers). With a few exceptions, all the seven genes had intron–exon junctions that matched the GT/AG boundary rules [43]. Furthermore, the phasing of the intron insertion, identified as the location of the intron before the first, second, or third nucleotide position of the neighbouring codon and referred to as phases 0, 1, and 2, respectively [44], seemed to be equally well conserved (Table 1). The high conserved genomic organization detected among the *P. laricio MBOS/MTPS* genes provided strong evidence of their common origin in conifers, confirming previous phylogenetic and protein structural studies, which demonstrated that the genes involved in the hemiterpene synthesis evolved independently in angiosperms and gymnosperms [6,12,34].

To gain insights into the functional evolution of terpene synthase genes in plants, Trapp and Croteau [41] divided them into three classes, namely, I, II, and III, which might have evolved sequentially by intron loss mechanisms. According to such a classification, the seven *P. laricio MBOS/MTPS* genes isolated in the present study belonged to class II, formed by gymnosperm monoterpene and sesquiterpene synthase genes containing nine introns. According to Trapp and Croteau [41], the gene coding for the bifunctional DTPS abietadiene synthase from *A. grandis* (*AgAs*) could represent the most obvious modern candidate that resembles the ancestral progenitor gene in gymnosperms.

To further complete the analysis of Trapp and Croteau [41], the genomic sequences of *P. laricio* MTPSs isolated in the present work were compared with those of the MTPSs already assigned to class II and with representative members of class I and III genes (Figure 4). Such a comparison indicated that, as already noticed among the seven *MBOS/MTP*S genes from *P. laricio* (see above), the number, position, and phase of the nine introns, numbered according to Trapp and Croteau [41] as III and VII-XIV, were highly conserved in all the class II *MTPS* genes. Moreover, the comparison of the isolated *P. laricio* pine genes and the available class II genes, with the class-I *AgAs* gene regarded as descending from a putative ancestral progenitor of all the *TPS* genes (see above), confirmed that class I genes gave rise to class II genes by a complex intron-loss mechanism. In particular, class II genes lost introns I and II and the entire N-terminal domain, named the conifer diterpene internal sequence (CDIS), spanning the regions of exons four, five, and six, and a small portion of exon seven, which included introns IV, V, and VI of all class I genes (Figure 4). In addition, by considering the comparison with the gene encoding for the limonene synthase in *Arabidopsis* (*AtLS*), it was confirmed that the class III genes, which included mono-, sesqui-, and diterpene synthase genes involved in the secondary metabolism in angiosperms, were derived from class II types by a further loss of intron VII and the sequential loss of introns IX and X (Figure 4). According to Trapp and Croteau [41], the class III genes contain the six conserved introns (III, VIII, XI-XIV) that are found in all *TPS* genes (Figure 4).

### 2.5. Transcript Profiling of P. laricio Monoterpene Synthase Genes Reveals Differential Expression across Different Tissues and Suggests Their Putative Roles in the Biosynthesis of Monoterpenes

All seven Calabrian pine *MBOS/MTPS* genes were shown to be constitutively expressed in all the five tissues studied, but their transcription levels were highly variable (Figure 5 and Appendix A).

Compared to the six *MTPS* genes, *Pnl MBOS1* was highly expressed in all five tissues studied, with the highest transcript levels detected in LS. The expression levels of such a gene were also comparatively high in YN, MN, and R, while a lower amount of the transcripts was detected in IS (Figure 5 and Appendix A). Based on the protein sequence homology (see above, Figure 2), the protein encoded by *Pnl MBOS1* may be involved in the synthesis of hemiterpenes, such as isoprene and/or its isomer MBO. Isoprene is known to be produced by many plant groups, especially Angiosperms trees, while in gymnosperms, this hemiterpene is known to be emitted from species belonging to the genus *Picea*, including *P. abies* and *P. glauca*, but not from species belonging to the genus *Pinus* [45,46]. Instead, the related hemiterpene MBO is emitted by *Pinus* species native to Northern America, e.g., *P. ponderosa*, *P. contorta,* and *P. jeffreyi* [34,47]. A quali/quantitative analysis of the hemiterpenes was not carried out in the five *P. laricio* tissues examined, because the identification and quantification of these compounds requires dedicated sampling equipment and analytical procedures, such as selected ion flow tube (SIFT)-MS [40], with respect to those used for the identification and analysis of the C_10_ monoterpenes. However, the comparatively high expression level detected in the present study for *Pnl MBOS1* suggested that the hemiterpenes could be produced in, and conceivably emitted by, different tissues of *P. laricio*. Based on these findings, further studies should be planned to elucidate the molecular and biochemical basis of hemiterpene formation in *P. laricio*.

Among the five tissues analysed, the relative amount of transcripts for each of the six *MTPS* genes was the highest in LS (Figure 5 and Appendix A), which was also the tissue that accumulated the largest amount of monoterpenes (Figure 1A). By considering all the five tissues, the highest level of expression was detected for *Pnl MTPS6* and *Pnl MTP5,* compared to the remaining four *MTPS* genes (Figure 5 and Appendix A).

In addition to LS, the expression levels of *Pnl MTPS6* were also comparatively high in MN and IS, and moderate in R and YN (Figure 5 and Appendix A). The tissue-specific expression pattern of *Pnl MTPS6,* found to be highly homologous to *P. contorta* and *P. banksiana MTPS* genes encoding β-phellandrene synthase (see Section 2.3, above), appeared to indeed be consistent with the tissue-specific levels of β-phellandrene, which was found to be the predominant monoterpene in three of the five tissues studied, namely, MN, IS, and YN (Figure 1B–F).

*Pnl MTPS5*, which based on sequence homology was predicted to produce α-pinene (see Section 2.3, above), was highly expressed in LS and MN tissues, followed by moderate levels of transcripts detected in YN and R, and low levels in IS (Figure 5 and Appendix A). A similar expression pattern was observed for *Pnl MTPS7*, whose predicted protein could also be involved in the synthesis of α-pinene (see Section 2.3, above), although its transcription levels were significantly lower than those of *Pnl MTPS5.* By matching the expression of the two genes with the corresponding monoterpene profiles, a remarkable level of correlation was found. Indeed, the highest amount of α-pinene was detected in LS, followed by the MN, YN, R, and IS tissues (Figure 1B–F).

The expression levels of *Pnl MTPS4* were comparatively high in LS and R in respect to the very low amount of transcripts detected in YN, MN, and IS (Figure 5 and Appendix A). Again, tissue-specific gene expression levels appeared to be consistent with the corresponding monoterpene profiles; indeed, the predicted protein sequence of *Pnl MTPS4* exhibited a high sequence identity with four proteins from *P. contorta* and *P. banksiana*, which were shown to produce β-pinene as their major product (see Section 2.3, above). As a matter of fact, compared to the other *MTPS* genes, *Pnl MTPS4* showed the highest level of expression in roots (R), in which β-pinene was the predominant monoterpene (Figure 1F).

Finally, *Pnl MTPS2* and *Pnl MTPS3* exhibited the lowest transcript levels among the six isolated *MTPS* genes in all the five tissues studied (Figure 5 and Appendix A). Based on sequence homology, the encoded proteins from *Pnl MTPS2* and *Pnl MTPS3* were predicted to produce α-terpineol and δ-3-carene, which were found to accumulate in very low amounts in the *P. laricio* tissues studied here (Figure 1B–F).

In summary, the transcript profiling of the *P. laricio MTPS* genes revealed a differential expression across the different tissues and were found to be consistent with the corresponding monoterpene profiles, suggesting potential roles for the six *MTPS* genes in the biosynthesis of the monoterpene compounds in this non-model species. However, because it is not always possible to predict the catalytic capability of the TPSs based only on the sequence similarity, the specific functions of the isolated genes need to be further verified. Further work is also needed concerning the expression analysis and transcript profiling of those monoterpenes, such as limonene, bornyl acetate, and α-tujene, which accumulated in comparatively low/moderate amounts in *P. laricio* tissues (see Figure 1B–F). On the one hand, in fact, the isolation of FL cDNAs encoding for less represented monoterpene components of conifer oleoresin is fraught with difficulties arising, first, from the scarcity of retrievable sequences in public databases. As a matter of fact, while monoterpene synthases producing limonene have been isolated from a few conifers, including Norway spruce (*P. abies*) [48], Sitka spruce (*P. sitchensis*) [49], and grand fir (*Abies grandis*) [50], those responsible for the biosynthesis of (+)-bornyl diphosphate, the likely precursor of bornyl acetate, and of α-tujene have been isolated and characterized in *Salvia officinalis* [51] and *Litsea cubeba* [52], respectively, but never in any conifer species. Secondly, those less represented monoterpenic components might have conceivably resulted from the transcription of low expression/under-represented genes, whose characterization is difficult on the background of a very complex genetic family as *MTPS* is in conifers.

This notwithstanding, studying those less represented monoterpenes at the molecular levels could be worthwhile, because some of them could undergo inductive dynamics in response to a wealth of environmental cues. As a matter of fact, in a previous work of ours conducted on adult individuals of *P. laricio* thriving in the same environment of the present study [27], we found that the foliar emissions of bornyl acetate and, to a lesser extent, of β-ocimene, differentiated plants infested by the caterpillars of the pine processionary moth (*Thaumetopoea pityocampa*) from their respective non-infested controls, being higher in the former during the period of maximal trophic activity of the larvae. On such basis, we are currently pursuing the isolation of FL cDNAs coding for the enzymes responsible for producing limonene, bornyl acetate, and α-tujene in *P. laricio*, to decipher their possible functional and ecological roles in plant–environment biotic and abiotic interactions.

## 3. Materials and Methods

### 3.1. Plant Material

The plant material, as well as the sampling procedure, were the same as those adopted in our previous work concerning the metabolic and genetic profiling of diterpenoids in different tissues of three-year-old seedlings of *P. laricio* (*Pinus nigra* J.F. Arnold subsp. *laricio* Palib. ex Maire) [28]. Briefly, the pine seedlings were grown in the open inside protective housing set up within the premises of the Aspromonte National Park, Southern Italy (38°17′27″ N, 15°81′68″ E; altitude 1010 MASL, exposed East). Five tissue types were collected for analysis: young needles (YN); mature needles (MN); bark and xylem combined from leader stem (LS); bark and xylem combined from interwhorl stem (IS); roots (R). All the collected tissues were immediately frozen in liquid nitrogen and stored at −80 °C until use for nucleic acid isolation and for metabolite profiling analysis.

### 3.2. Extraction and GC–MS Analysis of Monoterpene Metabolites

Approximately 500 mg of the five different tissues were extracted by sonication in 1 mL of n-pentane at room temperature for 15 min. In total, 100 μL of each extract was transferred into a conical vial and 0.5 μL was analysed by high-fast GC–MS techniques. For all the five tissue types, three biological replicates were processed, and then each of them analysed in triplicate.

Qualitative and quantitative GC–MS analyses of monoterpenes from *P. laricio* tissues were carried out essentially as reported in our previous work on diterpenes from the same plant source [28], in which full details are provided concerning the analytical equipment, the GC capillary column, the nature and flow rate of the carrier gas, the sample volume and the injection technique, the couped MS detector, and the temperatures selected for the transfer line, the ion source, and the analyser. For the present GC analysis of monoterpenes, the following thermal conditions were adopted: from 45 °C (1 min) to 250 °C (0.5 min) at 30 °C min^−1^, then to 325 at 45 °C min^−1^, then isothermal for 5 min. The MS acquisition was carried out under full scan (*m*/*z* 37–250) or selective ion monitoring mode at m/z: 136, 121 and 93, for qualitative and quantitative analyses, respectively.

Monoterpenes identification was achieved by comparing the experimental mass spectra both with those in the NIST08 and Wiley02 libraries and with those reported in the available literature [14,53]. As far as the Wiley and NIST mass spectra libraries were concerned, the spectral match scores obtained for the analysed monoterpenes in the pine tissues were invariably higher than 850, consistently returning the correct metabolite identification as the “first hit”. According to the NIST library guidelines, this match value is considered satisfactory and reliable for a correct identification of a given molecule.

The analyte concentrations, expressed as µg (g dry weight)^−1^, were calculated by calibration curves obtained by using commercial standards of α-pinene ((1*S*,5*S*)-2,6,6-trimethylbicyclo(3.1.1)hept-2-ene, 98% purity, Sigma-Aldrich, catalogue # 147524) and of (R)-(+)-limonene (1-methyl-4-(prop-1-en-2-yl)cyclohex-1-ene, 97% purity, Sigma-Aldrich, catalogue # 183164).

The GC–MS methods used in the present study for the extraction and analysis of plant metabolites were adequately validated for their selectivity, precision, and efficiency. Selectivity was verified by observing that no interfering peak was apparent at the elution time of each target analyte upon injecting three replicate blank samples. Precision was tested by measuring the inter- and intraday variability in the chromatographic profiles of spiked samples, which ranged from 2 to 7% in terms of relative standard deviation. Finally, the extraction method recovery was computed as the average of three replicate samples of the plant tissue spiked with a known aliquot of α-pinene and (R)-(+)-limonene standard mix and then analysed by GC–MS. Regardless of the tissue extracted, the measured mean recovery always ranged from 80 to 90%.

### 3.3. Isolation, Characterization, and Expression Analyses of Monoterpene Synthases Genes in P. laricio

As the first step, a multiple sequence alignment of the MTPS deduced proteins from the genus *Pinus* was carried out from which the corresponding phylogenetic tree was generated. The adopted in silico approach and the criteria used for phylogenetic analysis were described in detail in our previous work [6,28].

The genomic DNA and the total RNA were extracted from the five different tissue types of *P. laricio* as reported in Alicandri et al. [28]. All the DNA and cDNA samples were stored at −80 °C until used.

RT-polymerase chain reaction (PCR) was used to amplify partial cDNA coding for MTPSs by using forward and reverse primers designed in conserved regions among the *Pinus* MTPS sequences of the different groups identified by the phylogenetic analysis [6]. The complete list of the forward and reverse primers used is reported in Appendix A.

The partial cDNAs sequences obtained were then used as templates to isolate the corresponding full-length *MTPS* cDNAs by means of 5’ and 3’ RACE (Rapid Amplification of cDNA Ends) extensions, as detailed in our previous work [28]. The sequence of RACE primers used is reported in Appendix A.

To isolate the genomic MTPS sequences, the genomic DNA was amplified by using specific forward and reverse primers, designed in the close proximity of the initiation (ATG) or stop codons, respectively, of each full-length cDNA (Appendix A), as described in Alicandri et al. [28].

The cloning and sequencing of partial cDNAs, RACE, and genomic amplification products were conducted as described in Alicandri et al. [28]. Three different clones for each cDNA, genomic, and RACE amplicon were sequenced. In our same previous work, full details can be found concerning the analysis of the nucleotide sequences obtained and of the corresponding deduced amino acid sequences.

As far as the expression analysis of the isolated *MTPS* genes is concerned, full methodological details can be found in previous work of ours [28,54,55]. These include samples replication, quantitative real-time (qRT-PCR) conditions, the selection of the most appropriate and stable reference genes for normalization, the designing of primer pairs for both target and reference genes (Appendix A), the evaluation of primer specificity and amplification efficiency, and the criteria used to calculate normalized relative values of gene expression and their standard deviation.

### 3.4. Statistical Analysis

Each reported value for metabolites and gene expression levels represents the mean of a total of nine replicates, obtained from three biological replicates and three technical replicates for each biological replicate. The statistical significance of the differences observed was evaluated by one-way ANOVA, followed by the Tukey’s test. All statistical analyses were performed using JMP PRO 15 (Trial Version ©SAS Institute Inc., Cary, NC, USA).

## 4. Conclusions

The importance of terpenes in the physiological and ecological processes in plants and conifers, as well as their current and potential practical uses, deserve adequate consideration. In the present study, we carried out, for the first time to the best of our knowledge, a quantitative analysis of the monoterpene composition in different tissues of the non-model conifer species *Pinus nigra* subsp. *laricio* (*P. laricio*), namely, young needles, mature needles, the leader stem, interwhorl stem, and roots. In a bid to understand the molecular mechanism regulating terpene synthesis in the studied species, we also faithfully isolated and characterized monoterpene synthase genes in an attempt to predict their involvement in the specialized monoterpenoid metabolism.

All the *P. laricio* tissues examined indicated the presence of the same fourteen monoterpenes. It was shown that β-phellandrene was the most abundant monoterpene in the young and mature needles and in the interwhorl stem, while the leader stem and roots showed α-pinene and β-pinene, respectively, as the most abundant compounds. However, the leader stem revealed to contain the major quantity of monoterpenes among all the tested tissues. Taken together, these results indicated a tissue specificity in the quantitative composition of C_10_ terpenoids.

The phylogenetic analysis of the *Pinus* members of the terpene synthases d1-clade allowed the recognition of seven distinct groups. By examining the members of each phylogenetic group for their conserved regions, it was possible to design specific primers to be used for isolating from *P. laricio* seven FL transcripts, denoted as *Pnl MBOS1/Pnl MTPS2-7*, each belonging to one of the above phylogenetic groups. The subsequent analysis of the deduced amino acid sequences allowed to predict the potential roles of each of the *P. laricio* MTPSs in the synthesis of the monoterpenoids identified in this same species. The gene expression analysis revealed that the transcript profiling of the *P. laricio* MTPSs genes had differential abundances across the different tissues. Such tissue-specific expression profiles were found to be consistent with the corresponding monoterpene profiles, suggesting the potential involvement of the isolated *MTPS* genes in the biosynthesis of monoterpenoids.

Finally, the FL *MBOS/MTPS* cDNAs from *P. laricio* were the basis for isolating the corresponding complete genomic sequences, for each of which the exon/intron structure was determined. This filled a knowledge gap in the genomics of the *MTPSs* genes in *Pinus* spp., since no complete genomic sequence has been characterized so far in the non-model conifer species studied here.

The study of monoterpene synthase genes and of their putative functions in conifers appeared to be relevant because of the multiple functions that monoterpenes can provide, not only in terms of the physiological and ecological roles in plant fitness, but also for their proneness to be employed in a vast array of bio-based technological fields, including the biological control of plant pests and pathogens.

## Figures and Tables

**Figure 1 plants-11-00449-f001:**
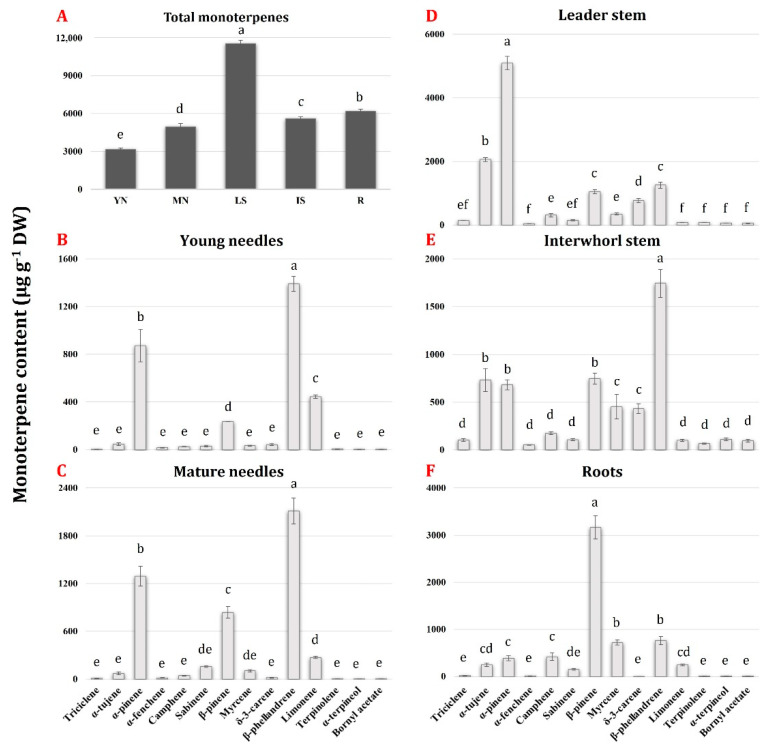
Total monoterpene contents (top left panel (**A**)) and levels of individual monoterpenes (panels (**B**–**F**)) in different tissues of three-year-old *P. laricio* saplings. Error bars indicate the standard deviation of the mean. The statistical significance of the differences was evaluated by one-way ANOVA, followed by a Tukey’s test. Different letters denote statistical significance of the difference at *p* < 0.01.

**Figure 2 plants-11-00449-f002:**
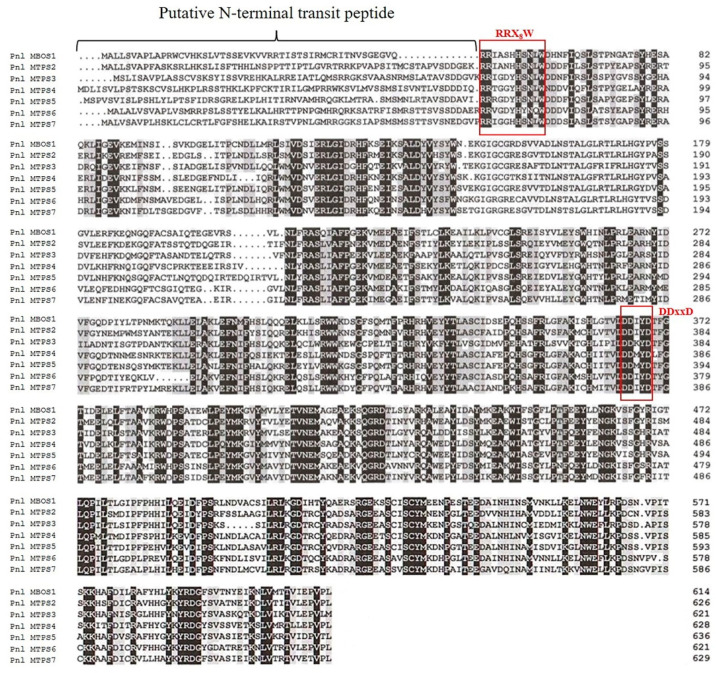
Alignment of deduced amino acid sequences of the seven putative hemi- and monoterpene synthases from *P. laricio* (Pnl MBOS1/Pnl MTPS 2–7) isolated in the present study. Amino acid residues with a black background indicate highly conserved regions, while amino acid residues which are identical in more than 50% of the proteins are in a grey background. The horizontal square bracket indicates the putative N-terminal transit peptide region. The “RRX8W” and class-I “DDxxD” signature motifs are indicated with red open rectangles.

**Figure 3 plants-11-00449-f003:**
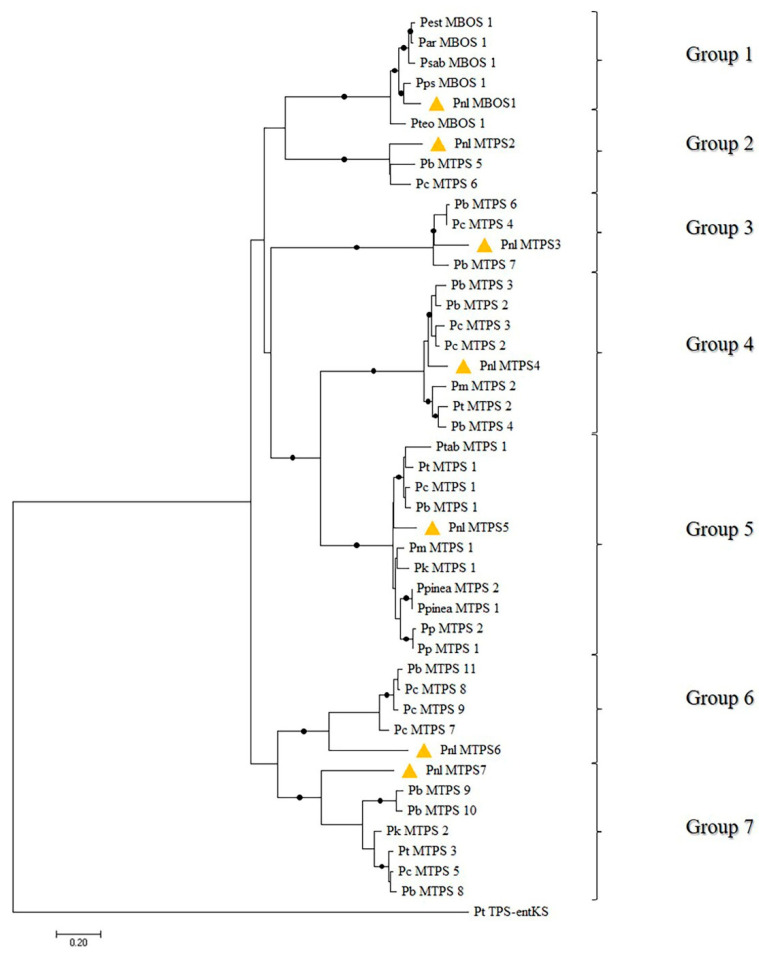
Phylogenetic tree of the deduced amino acid sequences of monoterpene synthase (*MTPS*) and 2-methyl-3-buten-2-ol synthase (*MBOS*) genes identified in different *Pinus* species (Appendix A) and those from the seven *P. laricio MBOS* and *MTPS* genes isolated in the present study (yellow triangles). The *ent*-kaurene synthase from *Physcomitrella patens* (Pt TPS-entKS, BAF61135) was used to root the tree. Branches marked with dots represent bootstrap support of more than 80% (1000 repetitions). The seven phylogenetic groups identified in the pine members of the TPS- d1- clade are indicated by square brackets. For the acronyms of the *Pinus* species, refer to Appendix A.

**Figure 4 plants-11-00449-f004:**
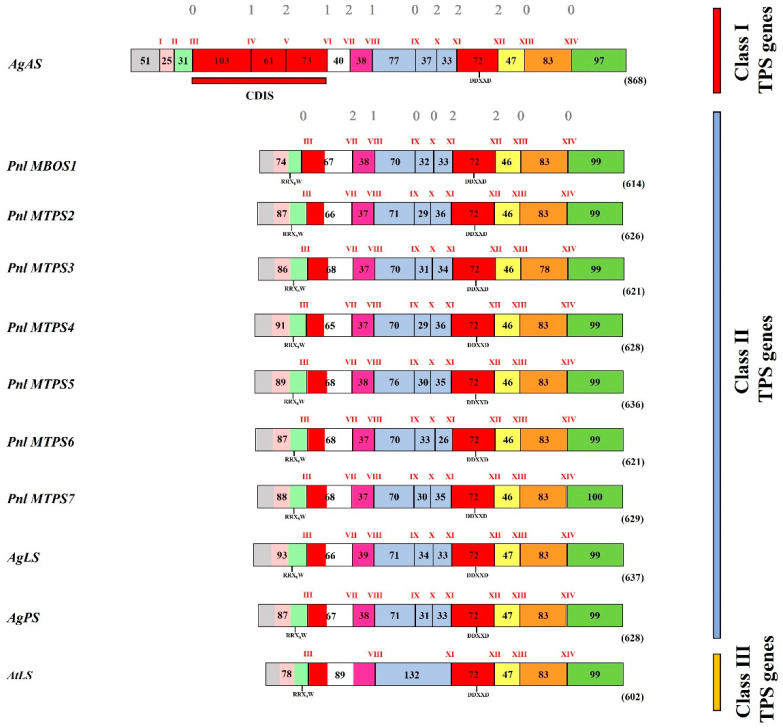
Genomic organization of class I, class II, and III terpene synthase genes sensu Trapp and Croteau [41]. Black vertical slashes represent introns (indicated by Roman numerals) and are separated among each other by coloured boxes with indicated lengths in amino acids, representing exons. The numbers above the introns of the first row from the top represent the intron phase type classification according to Li [44] and indicate conservation throughout the plants’ *TPS* genes. Schematization, intron numbers, and exon colouring scheme were based upon work by Trapp and Croteau [41]. Genomic DNA sequences compared were as follows: *AgAS*, *Abies grandis* abietadiene synthase (NCBI accession no. AF326516); *AgLS*, *Abies grandis* limonene synthase (AF326518); *AgLS, Abies grandis* pinene synthase (AF326517); *AtLS*, *Arabidopsis thaliana* putative limonene synthase (Z97341); *Pnl MBOS1/Pnl MTPS2-7,* the *MBOS* and *MTPS* genes isolated from *P. laricio* in the present study.

**Figure 5 plants-11-00449-f005:**
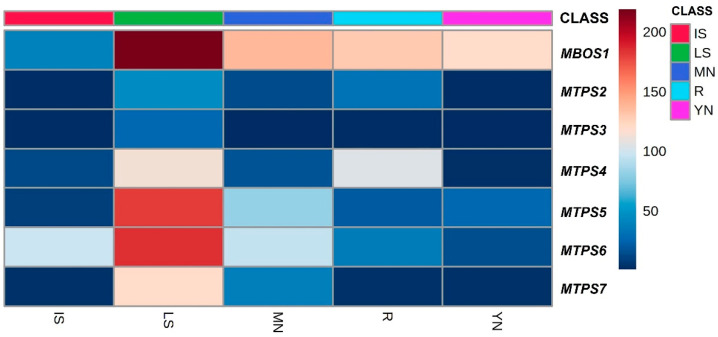
Heat maps of the relative expression levels of the hemiterpene synthase (*MBOS*) and of the monoterpene synthase (*MTPS*) genes isolated from five different tissues of *P. laricio*. The expression data of each gene were normalized using the geometric average of the two reference genes *CYP* and *upLOC*. Relative expression levels were calculated by setting a value of 1 for the lowest value among the seven genes in the five tissues considered (*MTPS3* in YN). For each gene, the differences in the relative expression levels were shown in colour according to the scale and statistical evaluation of the differences among the seven genes in the five tissues is reported in Appendix A. IS: bark and xylem combined from the interwhorl stems; LS: bark and xylem combined from the leader stem; MN: mature needles; R: roots; YN: young needles.

**Table 1 plants-11-00449-t001:** Comparison of the introns for the hemiterpene synthase (*MBOS*) and of the monoterpene synthase (*MTPS*) genes isolated in the present study: position (the letters refer to the last-coded amino acid of the exon), size (in parentheses), total number, and phase. No asterisk indicates no interruption between codons; single asterisk indicates intron inserted between the first and the second nucleotide; double asterisk indicates intron inserted between the second and third nucleotide; hyphen indicates intron not present. The introns are numbered according to approach by Trapp and Croteau [41].

Intron	*MBOS1*	*MTPS2*	*MTPS3*	*MTPS4*	*MTPS5*	*MTPS6*	*MTPS7*
**I**	-	-	-	-	-	-	-
**II**	-	-	-	-	-	-	-
**III**	G74 (97)	E87 (318)	G86 (177)	Y91 (102)	G89 (123)	E87 (91)	G88 (183)
**IV**	-	-	-	-	-	-	-
**V**	-	-	-	-	-	-	-
**VI**	-	-	-	-	-	-	-
**VII**	Y141 ** (159)	S153 ** (79)	R154 ** (81)	S156 ** (124)	S157 ** (90)	S155 ** (90)	S156 ** (89)
**VIII**	S179 * (115)	S190 * (101)	S191 * (106)	A193 * (148)	A195 * (80)	S192 * (111)	S193 * (104)
**IX**	E249 (94)	Q261 (225)	E261 (124)	E263 (263)	E271 (74)	E262 (97)	E263 (86)
**X**	L281 ** (269)	E290 ** (289)	S292 ** (186)	T292 ** (69)	E301 ** (181)	Q295 ** (221)	I293 ** (297)
**XI**	R314 ** (105)	R326 ** (108)	R326 ** (91)	R328 ** (125)	R336 ** (93)	R321 ** (94)	R328 ** (98)
**XII**	R386 ** (111)	R398 ** (111)	R398 ** (114)	R400 ** (91)	R408 ** (92)	R393 ** (98)	R400 ** (208)
**XIII**	A432 (99)	A444 (89)	A444 (100)	A446 (108)	A454 (111)	W440 (113)	A446 (89)
**XIV**	Q515 (94)	Q527 (112)	G543 (133)	Q529 (469)	Q537 (99)	A523 (112)	K529 (88)
**Genomic size (bp)**	2988	3313	2978	3386	2854	2893	3132
**Protein length (aa)**	614	626	621	628	636	621	629

## Data Availability

The data contained within the present article and in its Appendix A are freely available upon request to the corresponding author.

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
