# Peer review of "Monoterpene Synthase Genes and Monoterpene Profiles in Pinus nigra subsp. laricio"

_plants, 2022, doi:10.3390/plants11030449_

Round 1
Reviewer 1 Report
Dear Authors,
I have gone through the manuscript titled, "Monoterpene synthase genes and monoterpene profiles in Calabrian pine [Pinus nigra subsp. laricio (Poiret) Maire]", and find it very interesting to read.
This manuscript, which focuses on a poorly studied taxon, is in general well written, logically structured, well-illustrated and easy to understand. The title is confusing to me and needs to be changed (details below). The abstract is well written. It encapsulates the entire study (a bit of introduction, aim, result and outcome). However, if there the section could be shortened a bit to make it more concise, it would be great. The introduction is well written as it gives a good background of the research in question. Also, the aim of the study is evident in the beginning and concluding parts. I believe that the Materials and Methods section is well structured and scientifically sound. The figures and tables are correct.
Specific comments:
Title: I propose to shorten the title: Monoterpene synthase genes and monoterpene profiles in Pinus nigra subsp. laricio
I would like to emphasize that I have serious doubts as to what exactly the taxon was studied by the authors. The name "Calabrian pine" used by the authors is misleading, it refers to another taxon, i.e. Pinus brutia. This species - Pinus brutia, commonly known as the Turkish pine, is a species of pine native to the eastern Mediterranean region. Pinus brutia is also known by several other common names, including Calabrian pine, from a naturalized population of the pine in Calabria in southern Italy, from where the pine was first botanically described. The Corsican pine (Pinus nigra ssp. laricio) originates from the Mediterranean, where it thrives in dry warm conditions. It is a subspecies of the Austrian pine, Pinus nigra ssp. nigra, which has a rougher bark, is shorter in stature and is more branched in habit.
Maybe the authors were wrong? Maybe they meant a different name - the Corsican pine?
If I am not mistaken, the correct, accepted Latin name for this taxon is Pinus nigra subsp. laricio Palib. ex Maire not Pinus nigra subsp. laricio (Poiret) Maire - please verify it.
Lines 38-44: “Key to plant species:” - Is this paragraph necessary?
Line 82: “ … cleaning products.” - reference needed
Line 83: “ … additives” - reference needed
Lines 89-94: Sentence requires rephrasing
Line 142: Instead of common names, please add the Latin names of these taxa
Lines 557-560: Sentence requires rephrasing, please clarify
Lines 563-564: Better: Norway spruce (Picea abies), Sitka spruce (Picea sitchensis) and grand fir (Abies grandis) …
General: I strongly recommend separating the Results section from the Discussion section. It is not known what is the result of this project and which data has already been published. I have doubts the more that the authors have been studying the biology of Calabrian pine for a long time. They also published some articles in Plants (MDPI).
Author Response
Response to Reviewer 1 Comments
Point 1: The abstract is well written. It encapsulates the entire study (a bit of introduction, aim, result and outcome). However, if there the section could be shortened a bit to make it more concise, it would be great.
Response 1: Following the suggestion from the Reviewer 1, the abstract was modified from 250 to 172 words (Lines 15-27 of the present revised version).
Point 2: Title: I propose to shorten the title: Monoterpene synthase genes and monoterpene profiles in Pinus nigra subsp. laricio
Response 2: The suggested correction has been made in the title.
Point 3: I would like to emphasize that I have serious doubts as to what exactly the taxon was studied by the authors. The name "Calabrian pine" used by the authors is misleading, it refers to another taxon, i.e. Pinus brutia. This species - Pinus brutia, commonly known as the Turkish pine, is a species of pine native to the eastern Mediterranean region. Pinus brutia is also known by several other common names, including Calabrian pine, from a naturalized population of the pine in Calabria in southern Italy, from where the pine was first botanically described. The Corsican pine (Pinus nigra ssp. laricio) originates from the Mediterranean, where it thrives in dry warm conditions. It is a subspecies of the Austrian pine, Pinus nigra ssp. nigra, which has a rougher bark, is shorter in stature and is more branched in habit.
Maybe the authors were wrong? Maybe they meant a different name - the Corsican pine?
If I am not mistaken, the correct, accepted Latin name for this taxon is Pinus nigra subsp. laricio Palib. ex Maire not Pinus nigra subsp. laricio (Poiret) Maire - please verify it.
Response 3: We thank the Reviewer for the well aimed and learned criticism. As far as we know, the current accepted botanical classification of our plant material is "Pinus nigra J.F.Arnold subsp. laricio Palib. ex Maire" (= Pinus laricio Poir. subsp. calabrica (Loud.) Cesca & Peruzzi; = Pinus calabrica Hort. ex Gordon). Indeed, and unfortunately, from the standpoint of clarity and straightforwardness, such botanical entity is variously interpreted by the specialists, since certain of them regard it as a subspecies, some others as a variety, and still others even as a distinct species. So that, it appears that its botanical classification varies according to the appearance of studies concerning the taxonomy of the Pinus nigra group. In any case, in order to avoid generating further confusion in an already complex taxonomical issue, we eliminated in the present revised version, including the main text, tables headings and figures legends, as well as supplementary material, any reference to “Calabrian pine” and used only the currently accepted scientific name, i.e. Pinus nigra subsp. laricio, acronymized in the text, for the sake of briefness, as P. laricio.
Point 4: Lines 38-44: “Key to plant species:” - Is this paragraph necessary?
Response 4: Following the suggestion from the Reviewer 1, the paragraph “Key to plant species” has been removed.
Point 5: Line 82: “ … cleaning products.” - reference needed
Response 5: Done (Line 105 of the present revised version).
Point 6: Line 83: “ … additives” - reference needed
Response 6: Done (Line 107 of the present revised version).
Point 7: Lines 89-94: Sentence requires rephrasing
Response 7: Done (Lines 118-124 of the present revised version).
Point 8: Line 142: Instead of common names, please add the Latin names of these taxa
Response 8: Done (Line 179 of the present revised version).
Point 9: Lines 557-560: Sentence requires rephrasing, please clarify
Response 9: Following the suggestion of the Reviewer 1, the sentence was rephrased (Lines 586-599 of the present revised version).
Point 10: Lines 563-564: Better: Norway spruce (Picea abies), Sitka spruce (Picea sitchensis) and grand fir (Abies grandis) …
Response 10: Done (Lines 592-593 of the present revised version).
Point 11: General: I strongly recommend separating the Results section from the Discussion section. It is not known what is the result of this project and which data has already been published. I have doubts the more that the authors have been studying the biology of Calabrian pine for a long time. They also published some articles in Plants (MDPI).
Response 11: We thank the Reviewer for the above comment, even if we were unable to understand its meaning and implications in full. Supposing that the Reviewer was alluding to the need of a clearer separation between the results reported here from those obtained in previous studies of ours on the same subject, we do believe we made such distinction evident enough, by discussing our present results side by side with the previous ones in two different points of the Results and Discussion section, namely at the end of the 2.1 paragraph, for our previous study Alicandri et al. (2021), and at the end of the 2.5 paragraph, for our previous study Foti et al. (2020), with no apparent overlapping, neither confusion.
As for the Reviewer’s recommendation of separating the Results section from the Discussion one, we have to say that indeed our early draft was prepared by keeping the two sections separated from each other. Then, however, we changed our plans, because we realized that unifying the two made it easier, clearer and more concise the contextualization of the results on the background of the available literature, also keeping in mind the undeniable complexity of (mono)terpenoids metabolism and genetics. Thus, we would prefer to maintain our Results and Discussion joined together, even if we are of course ready to follow the Reviewer’s recommendation in case he/she maintain his/her request.
Reviewer 2 Report
The manuscript (plants-1571914) “Monoterpene synthase genes and monoterpene profiles in 2 Calabrian pine [Pinus nigra subsp. laricio (Poiret) Maire]” carried out a quantitative analysis of the monoterpenes composition in different tissues of the nonmodel conifer Pinus nigra subsp. Laricio and cloned and analyzed seven full length orthologous cDNA transcripts in Calabrian pine. This article is a continuation of the author's previous article. The research ideas are clear, the methods are appropriate, and the results are reliable, which has a certain research significance. But there is still something to pay attention to before the manuscript is accepted.
- The description of Fig. 3 in the result section is too redundant. I personally suggest merging descriptions.
- Fig. 4, the gene mark on the left should be in italics.
- Fig. 5, the gene mark on the left should be in italics.
- Line 211, Attention to the format of parentheses in literature citation!
- Line 133, Personal point of view, this will be better. ‘The tissue-specific and species-specific monoterpene metabolites in the Pinaceae’.
Author Response
Response to Reviewer 2 Comments
Point 1: The description of Fig. 3 in the result section is too redundant. I personally suggest merging descriptions.
Response 1: Following the suggestion from the Reviewer 2, the description of Fig. 3 has been merged (Lines 351-390 of the present version).
Point 2: Fig. 4, the gene mark on the left should be in italics.
Response 2: Done.
Point 3: Fig. 5, the gene mark on the left should be in italics.
Response 3: Done.
Point 4: Line 211, Attention to the format of parentheses in literature citation!
Response 4: Done (Line 247 of the present revised version).
Point 5: Line 133, Personal point of view, this will be better. ‘The tissue-specific and species-specific monoterpene metabolites in the Pinaceae’.
Response 5: Following the suggestion from the Reviewer 2, the subheading has been modified (Line 170 of the present revised version).
Reviewer 3 Report
The authors present a novel study on monoterpene synthase genes and monoterpene profiles in the Calabrian pine. Presented results (a quantitative analysis of monoterpene composition in different tissues and organs: namely young needles, mature needles, leader stem, interwhorl stem and roots) are new to the scientific community and worthy to publish. Methods are described in details.
Major suggestions:
However, could the authors discuss the problem of how infections (fungal, or insect) change monoterpene profiles ?
Did they check health condition of trees from which the material was collected?
As different organs were tested, could the authors link the differences in the monoterpene profiles with the number of the resin ducts in the individual organs?
Minor suggestion:
The Introduction section might be improved by using subheadings.
Author Response
Response to Reviewer 3 Comments
Point 1: However, could the authors discuss the problem of how infections (fungal, or insect) change monoterpene profiles?
Response 1: We thank the Reviewer for the above comment. On one side, in fact, we deliberately omitted from the original submission any reference to the involvement of monoterpenes in relation to plant biotic stress, first because the issue was perceived as not pertinent to the present manuscript, whose results were obtained by studying ostensibly healthy and juvenile plant material, certified, in terms of both genetic uniformity and phytosanitary status, by the official forest nursery of the Calabria Regional Authority, and what is more protected at least from insects attacks during the entire experiment (see Materials and Methods). The second reason for such omission was that, during the same experiment reported here, we indeed raised a parallel set of P. laricio saplings which were subjected to artificial infestation brought about by the lepidopteran Thaumetopoea pityocampa, popularly known as the pine processionary moth. The results of such parallel experiment, in which the plants described in the present manuscript acted as the healthy control, will be the subject of a separate manuscript, to be submitted in a short while. In any case, in order to encounter the Reviewer suggestion, we added to the present revised version the parts now appearing on Lines 60-85. We chose to add these parts to the Introduction, and not to the Result and Discussion section, in order to maintain pertinence and consistency (see above), although of course we are fully available for other options, if so requested by the Reviewer.
Point 2: Did they check health condition of trees from which the material was collected?
Response 2: We thank the Reviewer for the above comment. We did not run any specific check of the phytosanitary status of our plant material. We assumed its healthy conditions both relying upon the certification issued by the official forest nursery of the Calabria Regional Authority (the Calabria Verde Agency) and on the basis of permanent visual inspection carried out during the experiment by expert personnel, including a professional entomologist, belonging to the Calabria Regional Biodiversity Observatory, the Institution which hosted our experiment. Furthermore, we built an ad hoc protective housing (see the above comment), in which the P. laricio saplings were kept during the entire experiment, to prevent infestation from the most obvious phytofagous insect species, in particular the pine processionary moth, while ensuring full and permanent exposure of our material to the open field environment.
Point 3: As different organs were tested, could the authors link the differences in the monoterpene profiles with the number of the resin ducts in the individual organs?
Response 3: We thank the Reviewer for the comment. Unfortunately, we did not run any specific attempt to correlate monoterpene profiles with the number of resin ducts of the individual organs, although this would be a very interesting and important aspect to be considered in further work.
Point 4: The Introduction section might be improved by using subheadings.
Response 4: Following the suggestion from the Reviewer 3, the Introduction was modified in subheadings (Lines 32, 57, 110, 139).
Reviewer 4 Report
This study reports the biochemical diversity and genetic basis of monoterpenes in Calabrian Pine. The wider importance of monoterpenes in plants is well-introduced. The introduction flows well, moving from the roles and importance of terpenes in conifers, to the genes that synthesize them, and studies to investigate terpene diversity in conifers, to the study species. The different study aims, and how they are advances on previous work on this species, are clearly described. In the results and discussion section, comparisons with other studies added interest to the descriptions of monoterpene content in each tissue. Comparisons with diterpenes results previously reported by this group should be described as an aim at the end of the introduction. The phylogenetic results are well-linked to functional knowledge from the literature. Later phylogenetic results support the identification of these genes by comparing their characteristics with other genes in these families. Comparison of the expression results to the monoterpene profiles of each tissues add further insights to the function of these genes. The authors take care to stress that direct functional tests are required for verification. The results and discussion section concludes with consideration of some active follow up studies. The methods provide sufficient detail for repeatability and to confirm reliability of the results. The conclusions concisely summarize the results from the study and emphasize how they build knowledge of monoterpene diversity in conifers and their related physiological and ecological functions. Overall, i find this to be a thorough and carefully performed study that builds knowledge of the biochemical diversity of conifers.
Specific comments
L23-27 Break up this long statement to make it clearer.
L35 "genomic organization of plant terpene synthase genes" seems to me to be too long an expression to count as a keyword.
L46 I would add a simple definition of terpenoids early in the introduction for non-specialist readers.
L74 "host" can be dropped here.
L76-78 Incorporate this short paragraph into the first paragraph of the introduction.
L86 Perhaps here is a good place to introduce the basic chemical charactersitics of monoterpenes.
L131 Replace "confronted" with "compared"
Figure 1 The reason for testing for differences in content between different monoterpenes within the same tissue is not apparent to me. From the aims, it would be better to test for differences between tissues.
L204 Why use the term "quali-quantitatively" here? The GC/MS readings would seem quantitative to me.
L250 Specify the database that was BLAST searched here. State the criteria used for accepting a BLAST hit.
L260 State the methods used to produce the phylogenetic tree.
L265 The sequences would have already been aligned as part of the phylogenetic analysis. State the method used to align sequences.
L282 Drop "as" from "contain as many"
L284-287 I don't fully follow this result. Are you stating that the three clones had mostly identical sequence? rewrite for clarity.
L327 Should "proteins" be changed to "genes" here as DNA sequences are being compared?
Figure 3 State the units of the scale bar.
L498-499 Insert "with" between "[35], respect"
L672-674 See my earlier comment about the most appropriate unit of comparison being tissues rather than different monoterpenes within a tissue.
Author Response
Response to Reviewer 4 Comments
Point 1: Comparisons with diterpenes results previously reported by this group should be described as an aim at the end of the introduction.
Response 1: Following the suggestion from the Reviewer 4, this point has been added at the end of the introduction (Lines 164-168 of the present revised version).
Point 2: L23-27 Break up this long statement to make it clearer.
Response 2: Done. According also with the Reviewer 1, the abstract was modified to make it more concise and clearer (Lines 15-27 of the present revised version).
Point 3: L35 "genomic organization of plant terpene synthase genes" seems to me to be too long an expression to count as a keyword.
Response 3: Following the suggestion from the Reviewer 4, this keyword was modified in “genomic organization” (Lines 28-29 of the present revised version).
Point 4: L46 I would add a simple definition of terpenoids early in the introduction for non-specialist readers.
Response 4: Done (Lines 33-41 of the present revised version).
Point 5: L74 "host" can be dropped here.
Response 5: Done (Lines 99-100 of the present revised version).
Point 6: L76-78 Incorporate this short paragraph into the first paragraph of the introduction.
Response 6: Done (Lines 58-60 of the present revised version).
Point 7: L86 Perhaps here is a good place to introduce the basic chemical charactersitics of monoterpenes.
Response 7: Done (Lines 111-114 of the present revised version).
Point 8: L131 Replace "confronted" with "compared".
Response 8: Done (Line 163 of the present revised version).
Point 9: Figure 1 The reason for testing for differences in content between different monoterpenes within the same tissue is not apparent to me. From the aims, it would be better to test for differences between tissues.
Response 9: We thank the Reviewer for the comment, to which we reply that, on one side, we believe that adequate consideration was dedicated in our manuscript to the quantitative differences of monoterpenes as a group among the different tissues analyzed. On the other side, we have to add that the differences in content between different monoterpenes within the same tissue was also very important for us to assess. This is because we run, in parallel with the present experiment, another one in which we exposed a different set of P. laricio saplings to artificial infestation brought about by the lepidopteran Thaumetopoea pityocampa, popularly known as the pine processionary moth. In such context, we are of course very interested in understanding if, and how much, the monoterpene profiles (as well the expression of the corresponding genes) undergo changes following plant-host interactions, in each individual plant tissues, also depending on the feeding preferences and trophic temporal dynamics of the caterpillars. The results of such parallel experiment of ours, in which the plants described in the present manuscript acted as the healthy control, will be the subject of a separate manuscript, to be submitted in a short while.
Point 10: L204 Why use the term "quali-quantitatively" here? The GC/MS readings would seem quantitative to me.
Response 10: Done (Line 240 of the present revised version).
Point 11: L250 Specify the database that was BLAST searched here. State the criteria used for accepting a BLAST hit.
Response 11: We thank the Reviewer for the comment, to which we reply that in the first part of paragraph 2.2 we summarized some results reported in a previous paper of ours (Alicandri et al. On the evolution and functional diversity of terpene synthases in the Pinus species: A Review. J Mol Evol 2020, 88, 253–283. https://doi.org/10.1007/s00239-020-09930-8) to illustrate the approach used to isolate the P. laricio MTPS transcripts in the present study. All the procedures regarding the BLAST search, the phylogenetic analysis and the alignment of the amino acid sequences addressed in the paragraph 2.2 of the present manuscript, were reported in detail in the Supplementary file “Experimental procedure” of the aforementioned previous work of ours. In order to avoid the repetitions about the criteria of the aforementioned analyses, in the present work these methodologies are only mentioned.
Point 12: L260 State the methods used to produce the phylogenetic tree.
Response 12: We thank the Reviewer for the comment to which we hope we were able to reply convincingly in our previous response. All the procedures regarding the BLAST search, the phylogenetic analysis and the alignment of the aminoacid sequences were reported in detail in the Supplementary file “Experimental procedure” of our previous work: Alicandri et al. On the evolution and functional diversity of terpene synthases in the Pinus species: A Review. J Mol Evol 2020, 88, 253–283. https://doi.org/10.1007/s00239-020-09930-8. In order to avoid the repetitions about the criteria of the aforementioned analyses, in the present work these methodologies are only mentioned.
Point 13: L265 The sequences would have already been aligned as part of the phylogenetic analysis. State the method used to align sequences.
Response 13: We thank the Reviewer for the comment to which we hope we were able to reply convincingly in our previous responses on points 11 and 12. All the procedures regarding the BLAST search, the phylogenetic analysis and the alignment of the aminoacid sequences were reported in detail in the Supplementary file namely “Experimental procedure” of our previous work: Alicandri et al. On the evolution and functional diversity of terpene synthases in the Pinus species: A Review. J Mol Evol 2020, 88, 253–283. https://doi.org/10.1007/s00239-020-09930-8. In order to avoid the repetitions about the criteria of the aforementioned analyses, in the present work these methodologies are only mentioned.
Point 14: L282 Drop "as" from "contain as many"
Response 14: Done (Line 318 of the present revised version).
Point 15: L284-287 I don't fully follow this result. Are you stating that the three clones had mostly identical sequence? rewrite for clarity.
Response 15: We apologise for the lack of clarity of the results reported in L284-287 of the old version of the manuscript. According to the reviewer suggestion, the sentence in question was modified to make it clearer (Lines 320-324 in the revised version of the manuscript) and below is reported a brief explanation of the changes made.
Three different clones for each of the seven partial MBOS/MTPS transcripts and for each of the corresponding 5’ and 3’ RACE products were sequenced. For clarity, we added this sentence in the subheading 3.3 of the section Materials and Methods (Lines 685-686 of the revised version of the present manuscript). As described in the manuscript, in the case of the partial MTPS transcripts from groups 2, 4, and 7, two slightly different sequences were identified among the three clones analysed for each cDNA fragment. However, among the three sequenced clones for the 5’ and 3’ RACE products of each of the three partial MTPS transcripts we identified the same sequences, which were identical to the overlapping 5’ and 3’ regions of two of the three sequenced cDNA products for each of the three genes belonging to groups 2, 4 and 7. These results indicated that we were able to identify at least one of the two putative FL transcripts for groups 2, 4, and 7.
Point 16: L327 Should "proteins" be changed to "genes" here as DNA sequences are being compared?
Response 16: We thank the Reviewer for the comment. We propose to maintain the term “proteins” because the phylogenetic tree appearing in Figure 3 was built based on the deduced amino acidic sequences of the retrieved monoterpene synthase genes.
Point 17: Figure 3 State the units of the scale bar.
Response 17: We thank the Reviewer for the comment. The scale bar on the Figure 3 represents the measure of the phylogentic distance among the different sequences, and commonly it does not required of the units, but it’s rapresented with a decimal format.
Point 18: L498-499 Insert "with" between "[35], respect".
Response 18: Done (Lines 526-527 of the present revised version).
Point 19: L672-674 See my earlier comment about the most appropriate unit of comparison being tissues rather than different monoterpenes within a tissue.
Response 19: We thank the Reviewer for the comment, to which we hope we were able to reply convincingly in our previous comment concerning Figure 1, above.
Round 2
Reviewer 1 Report
The authors have adequately addressed my comments and suggestions towards improvement and/or correction. I have no further comments.